# The orchestrated cellular and molecular responses of the kidney to endotoxin define a precise sepsis timeline

Danielle Janosevic[1†], Jered Myslinski[1†], Thomas W McCarthy[1], Amy Zollman[1], Farooq Syed[2], Xiaoling Xuei[3], Hongyu Gao[3], Yun-Long Liu[3], Kimberly S Collins[1], Ying-Hua Cheng[1], Seth Winfree[1,4], Tarek M El-Achkar[1,4], Bernhard Maier[1], Ricardo Melo Ferreira[1], Michael T Eadon[1], Takashi Hato[1]*, Pierre C Dagher[1,4]*

[1]Department of Medicine, Indiana University School of Medicine, Indianapolis, United States; [2]Department of Pediatrics and the Herman B. Wells Center, Indiana University School of Medicine, Indianapolis, United States; [3]Department of Medical and Molecular Genetics, Indiana University School of Medicine, Indianapolis, United States; [4]Roudebush Indianapolis Veterans Affairs Medical Center, Indianapolis, United States

**Abstract** Sepsis is a dynamic state that progresses at variable rates and has life-threatening consequences. Staging patients along the sepsis timeline requires a thorough knowledge of the evolution of cellular and molecular events at the tissue level. Here, we investigated the kidney, an organ central to the pathophysiology of sepsis. Single-cell RNA-sequencing in a murine endotoxemia model revealed the involvement of various cell populations to be temporally organized and highly orchestrated. Endothelial and stromal cells were the first responders. At later time points, epithelial cells upregulated immune-related pathways while concomitantly downregulating physiological functions such as solute homeostasis. Sixteen hours after endotoxin, there was global cell–cell communication failure and organ shutdown. Despite this apparent organ paralysis, upstream regulatory analysis showed significant activity in pathways involved in healing and recovery. This rigorous spatial and temporal definition of murine endotoxemia will uncover precise biomarkers and targets that can help stage and treat human sepsis.

*For correspondence:
thato@iu.edu (TH);
pdaghe2@iu.edu (PCD)

†These authors contributed equally to this work

Competing interests: The authors declare that no competing interests exist.

## Introduction

Acute kidney injury (AKI) is a common complication of sepsis that doubles the mortality risk. In addition to failed homeostasis, kidney injury can contribute to multi-organ dysfunction through distant effects. Indeed, the injured kidney is a significant source of inflammatory chemokines, cytokines, and reactive oxygen species that can have both local as well as remote deleterious effects (*Husain-Syed et al., 2016*; *Lee et al., 2018*; *Deutschman and Tracey, 2014*; *Poston and Koyner, 2019*). Therefore, understanding the complex pathophysiology of kidney injury is crucial for the comprehensive treatment of sepsis and its complications.

We have recently shown that renal injury in endotoxemia progresses through multiple phases. These include an early inflammatory burst followed by a broad antiviral response that culminated in protein translation shutdown and organ failure (*Hato et al., 2019*). In a non-lethal and reversible model of endotoxemia, organ failure was followed by spontaneous recovery. The exact cellular and molecular contributors to this multifaceted response remain unknown. Indeed, the kidney is architecturally a highly complex organ in which epithelial, endothelial, immune, and stromal cells are at constant interplay. Therefore, we now examined the spatial and temporal progression of endotoxin injury to the kidney using single-cell RNA sequencing (scRNA-seq). Our data revealed that cell–cell

communication failure is a major contributor to organ dysfunction in endotoxemia. Remarkably, this phase of communication failure was also a transition point where recovery pathways were activated. We believe this spatially and temporally-anchored approach to the pathophysiology of endotoxemia is crucial for identifying potential sepsis biomarkers and therapeutic targets.

## Results

### ScRNA-seq identifies various renal cell populations

We harvested a cumulative amount of 63,287 renal cells obtained at 0, 1, 4, 16, 27, 36, and 48 hr after endotoxin (LPS) administration. The majority of renal epithelial, immune and endothelial cell types were represented (*Figure 1A*, https://connect.rstudio.iu.edu/content/18/). Note the absence of podocyte and mesangial cells, which can be a limitation of warm dissociation procedure (*Denisenko et al., 2020*). Cluster identities were assigned and grouped using known classical phenotypic markers (*Figure 1B*, *Figure 1—figure supplement 1A*; *Clark et al., 2019*; *Lake et al., 2019*; *Lee et al., 2015*; *Park et al., 2018*; *Ransick et al., 2019*). Interestingly, the UMAP-based computational layout of epithelial clusters recapitulated the normal tubular segmental order of the nephron. This indicates that cell type-defining gene expression patterns gradually change among neighboring tubular segments along the nephron. The expression of cluster-defining markers varied significantly during the injury and recovery phases of endotoxemia (*Figure 1—figure supplement 1B*; *Supplementary file 1*). Therefore, we also identified a set of genes that are conserved across time for a given cell type (*Figure 1—figure supplement 1C*).

In the integrated UMAP (*Figure 1A*), we noted the presence of a proliferative cell cluster (*Cdk1* and *MKi67* expression). The stress markers *Jun* and *Fos* that are typically associated with the dissociation procedure (*van den Brink et al., 2017*; *Denisenko et al., 2020*) were not strongly expressed in this proliferating cell cluster (*Figure 1—figure supplement 2A*). By back mapping to time-specific unintegrated UMAPs, we determined that these proliferating cells could be traced to specific cell types at various points along the endotoxemia timeline (*Figure 1C*). At baseline, proliferating indices localized to the proximal tubular cluster in uninjured tissues (*Figure 1C*). This was confirmed microscopically after in vivo thymidine analog injection (*Figure 1—figure supplement 2B*). Within the first hour after LPS, these proliferative indices were expressed primarily in S1 cells. These cells are the site of LPS uptake in the kidney as we have previously shown (*Hato et al., 2015*; *Hato et al., 2018*; *Kalakeche et al., 2011*). At later time points, proliferative indices are seen in lymphocytes (16 hr) and S3 cells (36 hr) (*Figure 1C*). The migration of proliferation indices among various cell types highlights the spatial and temporal nature of the renal response to LPS. These proliferative indices likely reflect cell cycle activity which may be involved in injury, repair or recovery processes (*Yang et al., 2010*).

### Integration of scRNA-seq and spatial transcriptomics localizes subtypes of S3 proximal tubules

Among the proximal tubular cells, we noted the presence of a distinct cluster expressing Angiotensinogen (Agt) and other unique identifiers such as *Rnf24*, *Slc22a7*, and *Slc22a13* (*Figure 2A*). This is likely the proximal tubular S3-Type 2 (S3T2) reported by others (*Cao et al., 2018*; *Ransick et al., 2019*). This cluster maintained a separate and distinct identity throughout most of the endotoxemia timeline (*Figure 1C*). Because the location of S3T2 is currently unknown, we performed in-situ spatial transcriptomics on endotoxemic mouse kidneys (*Ståhl et al., 2016*). We then integrated our scRNA-seq with the in-situ RNA-seq to map our scRNA-seq clusters onto the tissue (*Figure 2—figure supplement 1A and B*). We found that the S3 cluster localizes to the cortex while S3T2 is in the outer stripe of the outer medulla (OS-OM; *Figure 2B*, *Figure 2—figure supplement 1B*). We confirmed the location of S3T2 to the OS-OM with single-molecular FISH (*Figure 2—figure supplement 1C*). The differential gene expression between S3 and S3T2 is likely dictated by regional differences in the microenvironments of the cortex and the outer stripe. The small number of cells in this S3T2 cluster in our scRNA-seq data may be the result of a cortical dissociation bias. Finally, because angiotensinogen (*Agt*) was strongly expressed in S3T2, we also examined the expression of other components of the renin-angiotensin system as shown in *Figure 2—figure supplement 2*.

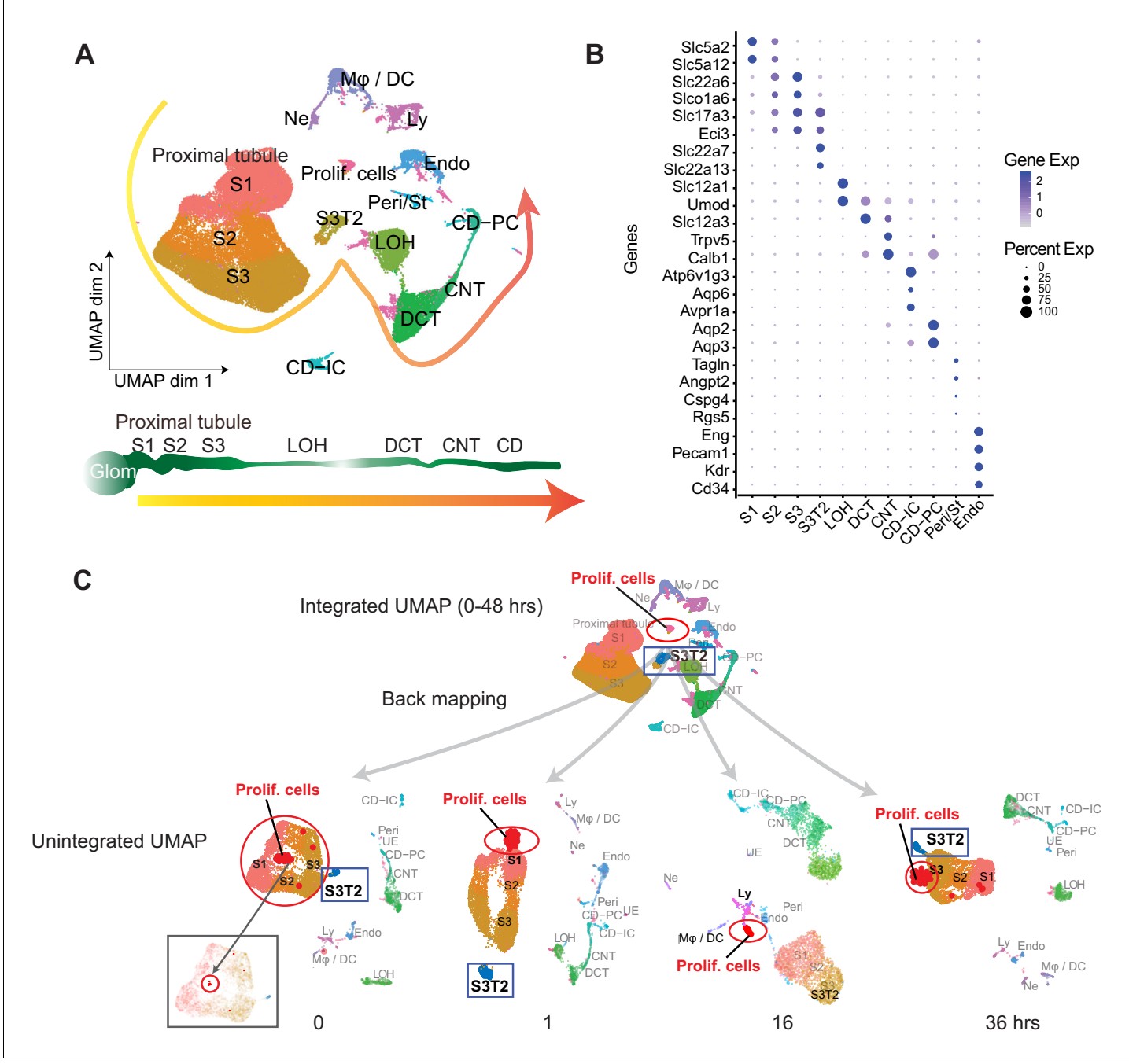

**Figure 1.** ScRNA-seq identifies various renal cell populations. (**A**) Integrated UMAP of kidney cell clusters from control and LPS-treated mice (0, 1, 4, 16, 27, 36, and 48 hr after LPS injection). Actual anatomical layout of kidney nephronal segments is shown below UMAP. (**B**) Dot plot of representative genes defining indicated cell types. (**C**) Backmapping of cells from the integrated UMAP onto unintegrated UMAPs of select time points. Highlighted are the proliferating cell cluster (red circle) and S3T2 cluster (blue box). For visibility the actual size of the proliferating cells cluster was purposefully exaggerated. Inset at the 0 hr timepoint shows the proximal tubular clusters with the non-edited proliferating cells subcluster circled in red. CD, collecting duct. CD-IC, collecting duct-intercalated cells. CD-PC, collecting duct-principle cells. CNT, connecting tubule. DCT, distal convoluted tubule. Endo, endothelial cells. Exp, expression. Glom, glomerulus. LOH, Loop of Henle. LPS, endotoxin. Ly, lymphocytes. Mφ -DC, macrophage-dendritic cells. Ne, neutrophil. Peri/St, mixed pericyte and stromal cells. Prolif. Cells, proliferating cells. PT, proximal tubule. S1, first segment of PT. S2, second segment of PT. S3, third segment of PT. S3T2, S3 type 2 cells.

The online version of this article includes the following figure supplement(s) for figure 1:

**Figure supplement 1.** Cluster-defining markers across the endotoxemia timeline.

**Figure supplement 2.** Characterization of proliferating cells.

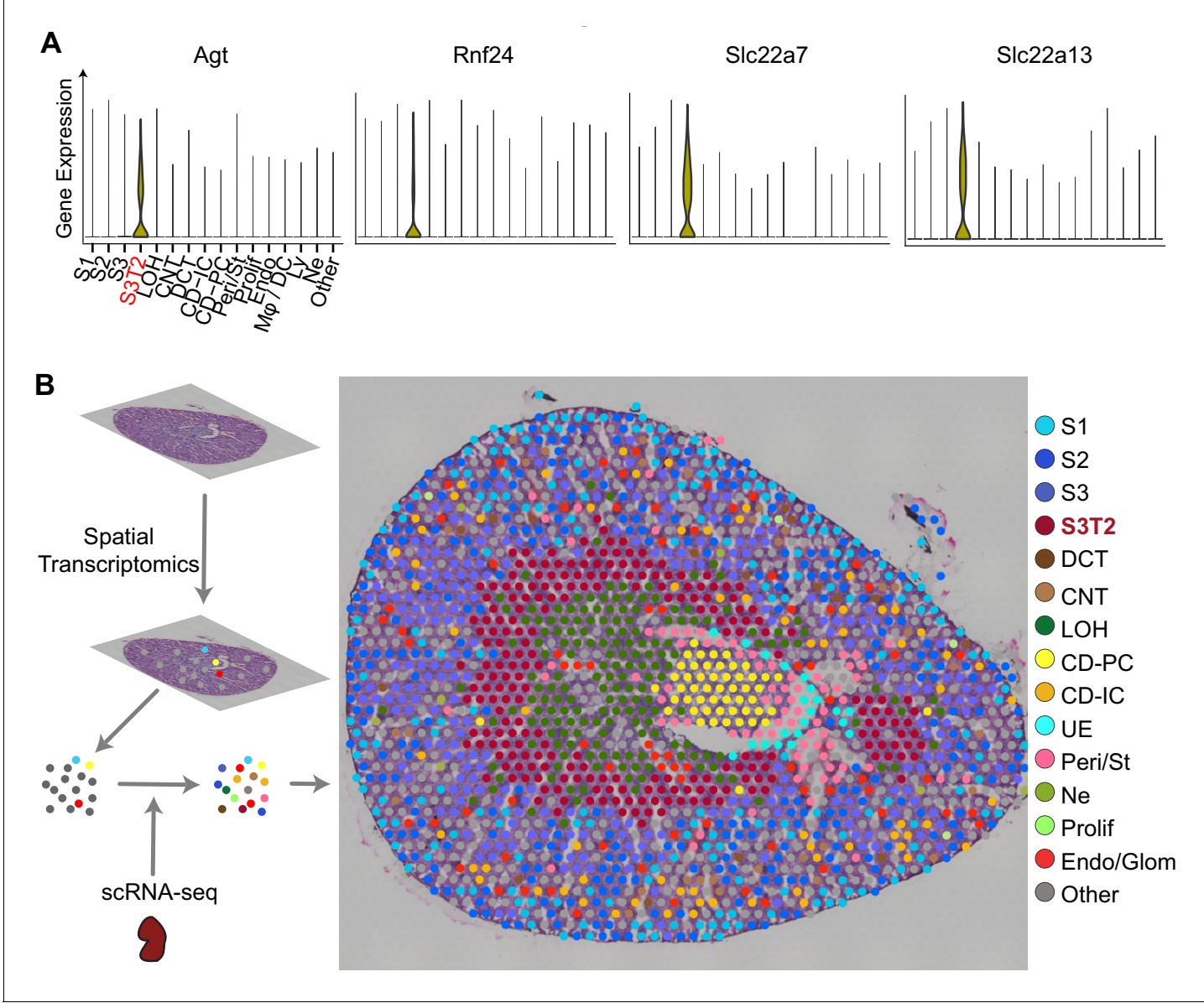

**Figure 2.** Integration of scRNA-seq and spatial transcriptomics localizes subtypes of S3 proximal tubules. (**A**) Violin plots of S3T2 defining markers. (**B**) Integration of spatial transcriptomics and scRNA-seq. Spatial transcriptomics were performed on a slice of mouse kidney. This yielded seven clusters that were expanded to 15 cell types by integrating spatial transcriptomics with scRNAseq data from LPS-treated mice. See also Figure S3.

The online version of this article includes the following figure supplement(s) for figure 2:

**Figure supplement 1.** Spatial transcriptomics validation.

**Figure supplement 2.** Cellular expression of the RAS axis along the endotoxemia timeline.

## Cell trajectory and velocity field analyses of scRNA-seq characterize subpopulations of immune cells

The immune cell content of the septic kidney was time-dependent and showed a five-fold increase in immune cells over 48 hr after LPS, primarily macrophages (*Figure 3A and B*). We noted two distinct macrophage clusters denoted as Macrophage A and Macrophage B (Mφ-A, Mφ-B). Both of these clusters expressed classical macrophage markers such as Cd11b (*Itgam*) (*Figure 3C*). However, they differed in the expression of *Adgre1* (F4/80, Mφ-A) and *Ccr2* (Mφ-B). The accumulated macrophages were predominantly Mφ-A. We noted the absence of proliferation markers (*Cdk1, MKi67*) in this cluster, raising the possibility that this may be an infiltrative macrophage type (*Figure 3D*). The

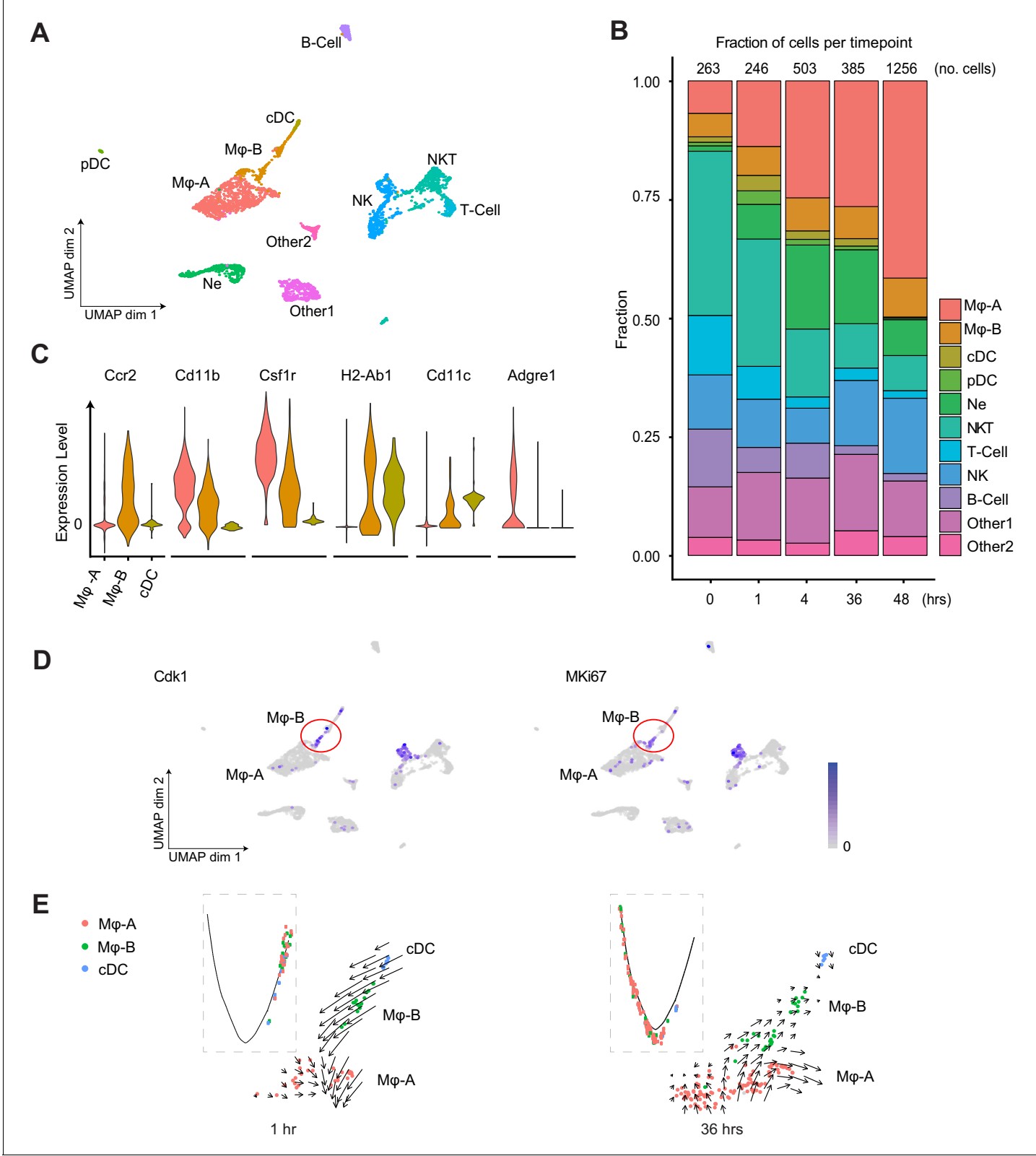

**Figure 3.** Endotoxemia induces dynamic changes in renal immune cell composition, pseudotime states and RNA velocity. (A) Integrated UMAP of the immune cell clusters from control and LPS-treated mice (0, 1, 4, 16, 27, 36, and 48 hr after LPS injection). Other one and Other two are Cd45+ cells with mixed epithelial and immune markers. (B) Stacked bar plot with fractions of immune cells (relative to total number of cells) shown in the y-axis, at 0, 1, 4, 36, and 48 hr after LPS. The total number of immune cells is indicated at the top of the bar for each time point. (C) Integrated violin plots from all time

*Figure 3 continued on next page*

*Figure 3 continued*

points for indicated genes defining subtypes of macrophages and DCs. (D) Feature plots of proliferation markers expression from integrated time points in the immune cell subsets. (E) Integrated cell trajectory analyses and RNA velocity fields for macrophages and dendritic cells shown at indicated time points. cDC, conventional dendritic cell. Hrs, hours. Mφ-A, macrophage-A. Mφ-B, macrophage-B. Ne, neutrophil. NK, natural killer cells. NKT, natural killer T-cells. pDC, plasmacytoid dendritic cell. T-cell, Cd3+ T-lymphocytes.

The online version of this article includes the following figure supplement(s) for figure 3:

**Figure supplement 1.** Immune cell subset characteristics.

Mφ-B cluster, located in the UMAP between Mφ-A and conventional dendritic cells (cDC) expressed also cDC markers such as MHC-II subunit genes (*H2-Ab1*) and Cd11c (*Itgax*) indicating that it is an intermediary macrophage type (*Figure 3C*). This continuum between macrophages and dendritic cells in the kidney has been reported (*Krüger et al., 2004*; *Woltman et al., 2007*; *Huen and Cantley, 2017*; *Gottschalk and Kurts, 2015*). Interestingly, Mφ-B cells expressed proliferation markers (*Cdk1, MKi67*) and thus, may be differentiating toward a Mφ-A or cDC phenotype (*Figure 3D*). Pseudotime and velocity field analysis suggested that at earlier time points (1 hr) Mφ-B was differentiating toward Mφ-A phenotype. At later time points (36 hr) the velocity field suggested that Mφ-B was differentiating toward cDC but pseudotime analysis was inconclusive (*Figure 3E*). Similarly, the Mφ-A cluster showed two subclusters on the RNA velocity map (*Figure 3—figure supplement 1A*). One of the subclusters showed increased expression of alternatively activated macrophage (M2) markers such as *Arg1* (Arginase 1) and *Mrc1* (Cd206) (*Lee et al., 2011*) at later time points (36 hr, *Figure 3—figure supplement 1B*). Therefore, RNA velocity analysis may be a useful tool in distinguishing macrophage subtypes in scRNA-seq data.

In T-cells, while *Cd4* expression was minimal at all time points, the expression of *Cd8* was robust and relatively preserved over time (*Figure 3—figure supplement 1C*). We also noted an increase in a distinct plasmacytoid dendritic cell cluster at 1 hr (pDC). These pDCs, along with natural killer (NK) cells, are known to signal through the interferon-gamma pathway and stimulate Cd8 expression (*Guillerey et al., 2012*; *Hemann et al., 2019*). This supports the early antiviral response we have previously reported in this endotoxemia model (*Hato et al., 2019*).

## Pseudotime and velocity field analyses identify cell-specific phenotypic changes along the endotoxemia timeline

We next examined the phenotypic changes in epithelial and endothelial populations along the enodotoxemia timeline (*Figure 4A*). At each time point, cells exhibited various states of gene expression that are well defined with pseudotime analysis. In fact, pseudotime accurately predicted future states as observed in real time. The endothelium exhibited changes in states as early as 1 hr, while S1 showed changes at later time points (4 hr). Importantly, the 16 hr timepoint was a turning point after which most cells returned to their baseline states indicating recovery (*Figure 4A*). At later time points, many cell types lost function-defining markers while acquiring novel ones. For example, S1 and S3 lost classical markers like *Slc5a2* (SGLT2) and *Aqp1* and expressed new genes involved in antigen presentation such as *H2-Ab1* (MHC-II) and *Cd74* (*Figure 4B*, *Figure 4—figure supplement 1A and B*). Similar changes were reported by others in bulk kidney RNA (*Tran et al., 2011*; *Figure 4—figure supplement 2*). Moreover, the highly distinct phenotypes that differentiated S1 from S2/S3 at baseline merged into one phenotype for all three sub-segments by 16 hr after LPS (*Figure 4C*). However, despite the apparent convergent phenotype at 16 hr, additional analytical approaches such as RNA velocity revealed significant differences in RNA splicing kinetics between S1 and S3 segments at this time point. In addition, RNA velocity revealed the presence of two subclusters within the S3 segment at 16 hr (*Figure 4D*). These two velocity subclusters did not correlate with the two states seen in pseudotime analysis. Overall, the distinct changes observed in the RNA velocity field point to the significance of altered splicing activities at 16 hr. This is also supported by our nascent proteomics analysis in which molecules involved in RNA splicing were overrepresented at this time point (*Hato et al., 2019*).

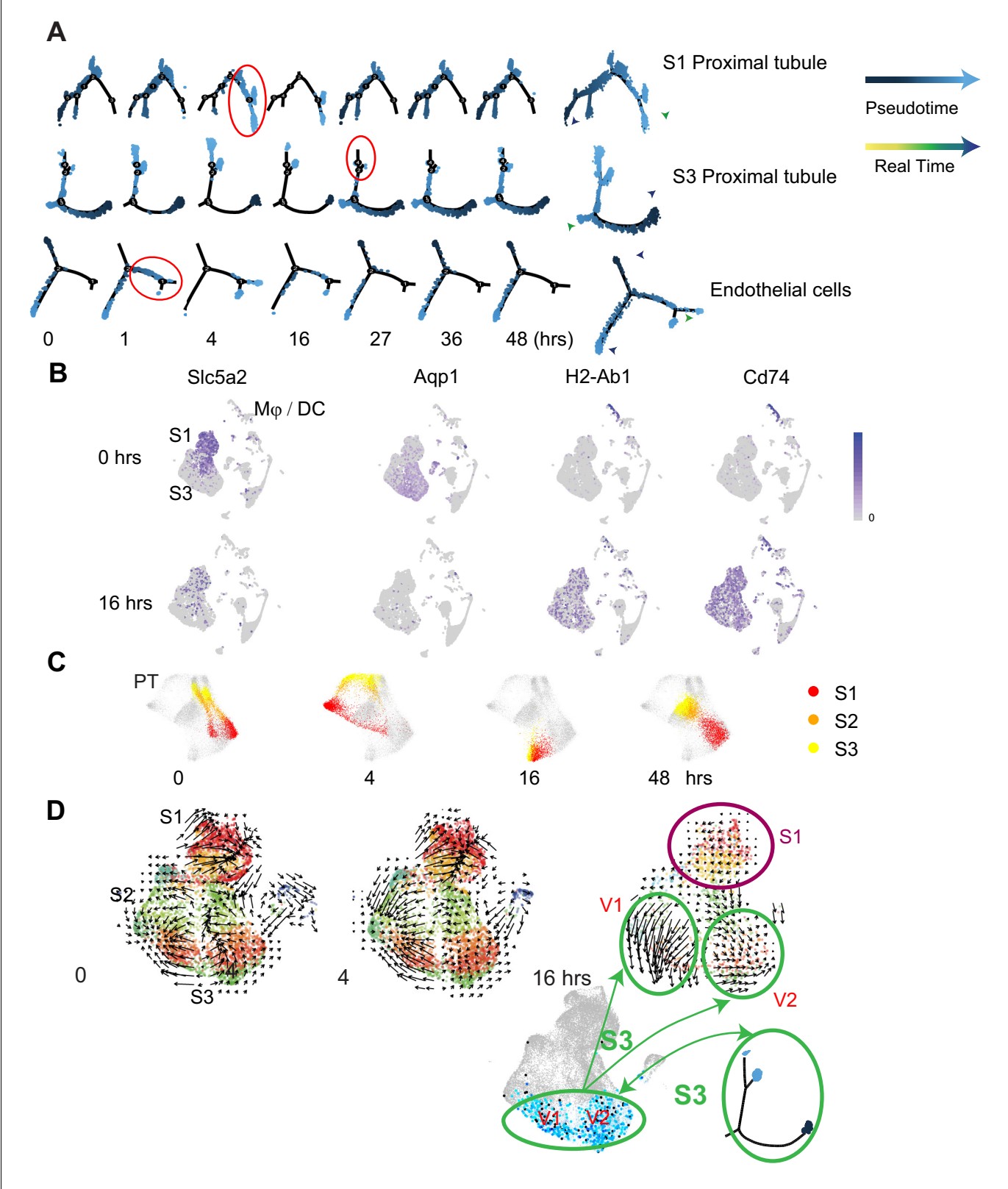

**Figure 4.** Pseudotime and velocity field analyses identify cell-specific phenotypic changes along the endotoxemia timeline. (**A**) Cell trajectory analysis for S1, S3, and endothelial cells shown at indicated time points. Highlighted in red circles are significant state transitions in respective cell types. The last cell trajectory shown for each cell type is integrated from all time points. It highlights the correspondence between pseudotime and real time. (**B**)

*Figure 4 continued on next page*

*Figure 4 continued*

Feature plots of selectgenes shown at indicated time points highlighting proximal tubular phenotypic changes. (C) Time-specific S1, S2, and S3PT cells (red, orange, yellow) overlaid on composite t-SNE map of all PT cells (gray). (D) RNA velocity fields for S1, S2, and S3 proximal tubular cells are shown at indicated time points. Two velocity subfields V1 and V2 in S3 cells are circled in green. Projections of two pseudotime S3 states (light blue, dark blue dots) onto the S3 velocity fields do not show a 1:1 correspondence with the two velocity subfields V1 and V2. V1, velocity subfield 1. V2, velocity subfield 2.

The online version of this article includes the following figure supplement(s) for figure 4:

**Figure supplement 1.** Transporter gene expression across the nephron during endotoxemia.

**Figure supplement 2.** Similarity analysis between bulk and single-cell RNA-seq data.

## Endotoxemia induces an organ-wide host defense phenotype in the kidney

Within 1 hr of LPS exposure, most cell types showed decreased expression of select genes involved in ribosomal function, translation and mitochondrial processes such as *Eef2* and *Rpl* genes (*Figure 5A*, *Figure 5—figure supplement 1A*). This reduction peaked at 16 hr and recovered by 27 hr. Concomitantly, most cell types exhibited increased expression of several genes involved in inflammatory and antiviral responses such as *Tnfsf9*, *Cxcl1*, *Ifit1*, and *Irf7*. However, this increase was not synchronized among all cell populations. Indeed, it occurred as early as 1 hr in endothelial cells, macrophages and pericyte/stromal cells, all acting as first responders. In contrast, epithelial cells were late responders, with increases in inflammatory and antiviral responses occurring between 4 and 16 hr. Importantly, 4 hr after LPS administration, cluster-specific Gene Ontology terms were indistinguishable among the majority of cell types with enrichment in terms related to defense, immune and bacterium responses (*Figure 5B*). One noted exception was the S3T2 cells (outer stripe S3) which did not enrich as robustly as other cell types in these terms. It mostly maintained an expression profile related to ribosomes, translation and drug transport throughout the endotoxemia timeline (*Figure 5—figure supplement 2*). Other players of interest in sepsis pathophysiology such as prostaglandin and coagulation factors are described in *Figure 5—figure supplement 1B*. We found that multiple cell types including epithelial, endothelial, immune and stromal cells contribute to the flow of these pathways at baseline and after injury. Because NF-κB is a major upstream transcription factor in the TLR4-endotoxin pathway, we show its spatial and temporal expression in the kidney by immunohistochemistry (*Figure 5—figure supplement 3*). As we previously reported (*Hato et al., 2019*), activation of NF-κB as evidenced by nuclear translocation was maximal in most tubules and interstitial cells/endothelial cells 1 hr after endotoxin.

At the 48 hr time point, while S1 cells mostly recovered to baseline, the macrophages showed increased expression of genes involved in phagocytosis, cell motility and leukotrienes, broadly representative of activated macrophages (e.g. *Csf1r*, *Lst1*, *Capzb*, *S100a4*, *Cotl1*, *Alox5ap*; *Figure 5A*). Intriguingly, at this late time point, the pericyte/stromal cells are enriched in unique terms related to specific leukocyte and immune cell types such as lymphocyte-mediated immunity, T-cell mediated cytotoxicity and antigen processing and presentation (*Figure 5B*). This suggests that the pericyte may function as a transducer between epithelia and other immune cells.

## Endotoxemia alters cellular crosstalk causing time-specific global communication failure

We next examined comprehensively cell–cell communication along the endotoxemia timeline (*Figure 6*, *Figure 6—figure supplement 1*, https://connect.rstudio.iu.edu/content/19/; the full list is available in *Supplementary file 2*). We show select examples of cell type-specific receptor–ligand pairs known to be involved in sepsis physiology (*Higgins et al., 2018*). For example, we found that S1 and endothelial cells communicate with the *Angpt1* (Angiopoetin 1) and *Tek* (Tie2) ligand-receptor pair at baseline and throughout the endotoxemia timeline (*Figure 6C*). In contrast, *C3* was strongly expressed in pericyte/stromal cells, while its receptor *C3ar1* localized to macrophage/DCs. This communication, present at baseline, did increase with time with additional players such as S1 participating in the cross talk (*Figure 6—figure supplement 1A*). Another strong communication was noted between endothelial cells and macrophage/lymphocytes using the *Ccl2* and *Ccr2* receptor–ligand pair. The architectural layout of these four cell types, with pericytes and endothelial cells

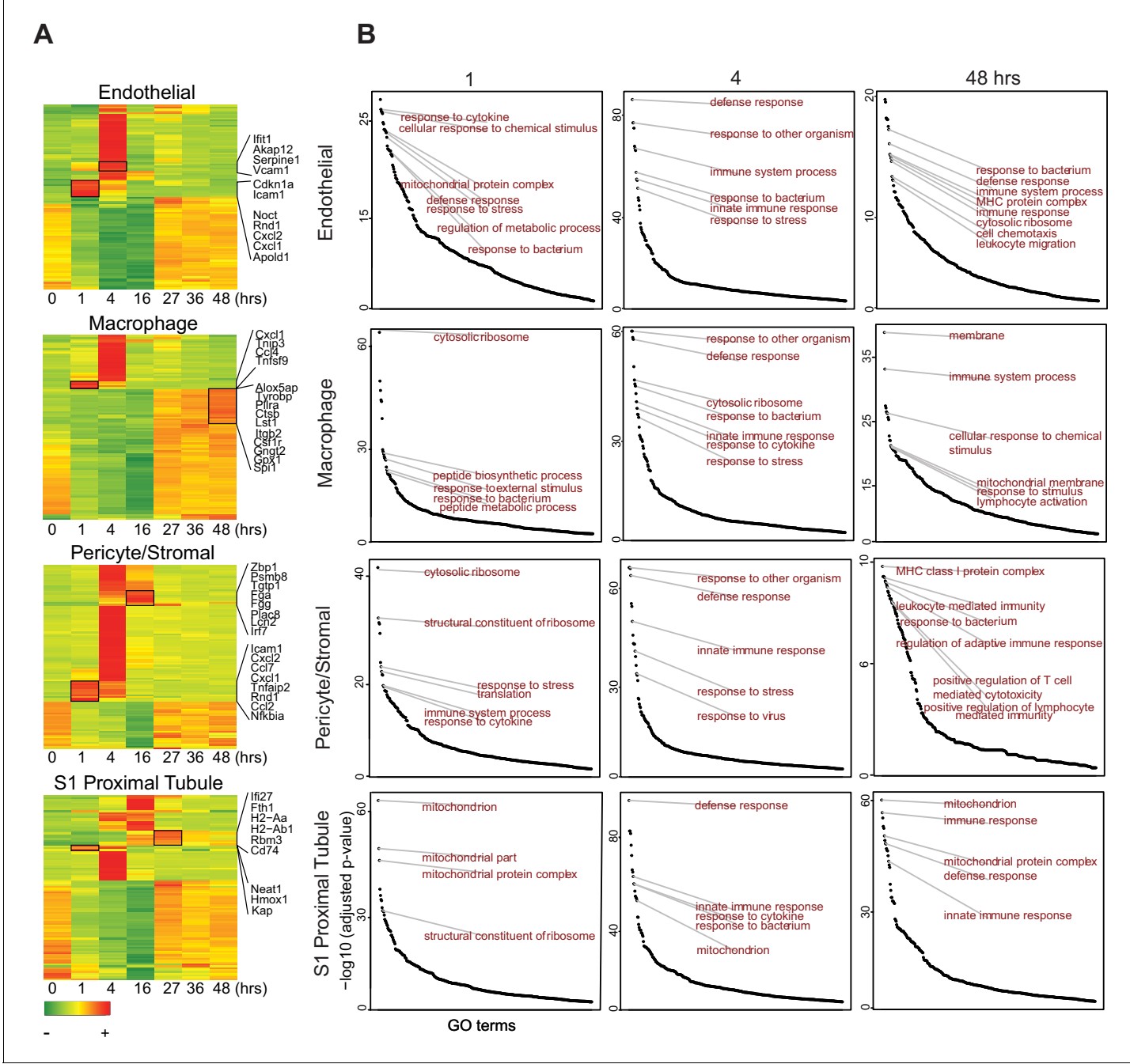

**Figure 5.** Endotoxemia induces an organ-wide host defense phenotype in the kidney. (**A**) Heatmaps of select cell types with top 100 differentially expressed genes across the endottoxemia timeline (0–48 hr). Select genes are shown for each cell type. (**B**) Time dependent enrichment of gene ontology terms for indicated cell types. GO terms are sorted in order of statistical significance. Hrs, hours. GO, gene ontology biological processes. The online version of this article includes the following figure supplement(s) for figure 5:

**Figure supplement 1.** Comparisons of transcriptomic profiles of selected cell types across the endotoxemia timeline.
**Figure supplement 2.** S3T2 GO terms.
**Figure supplement 3.** Representative images of immunohistochemical staining for NF-κB are shown under indicated conditions.

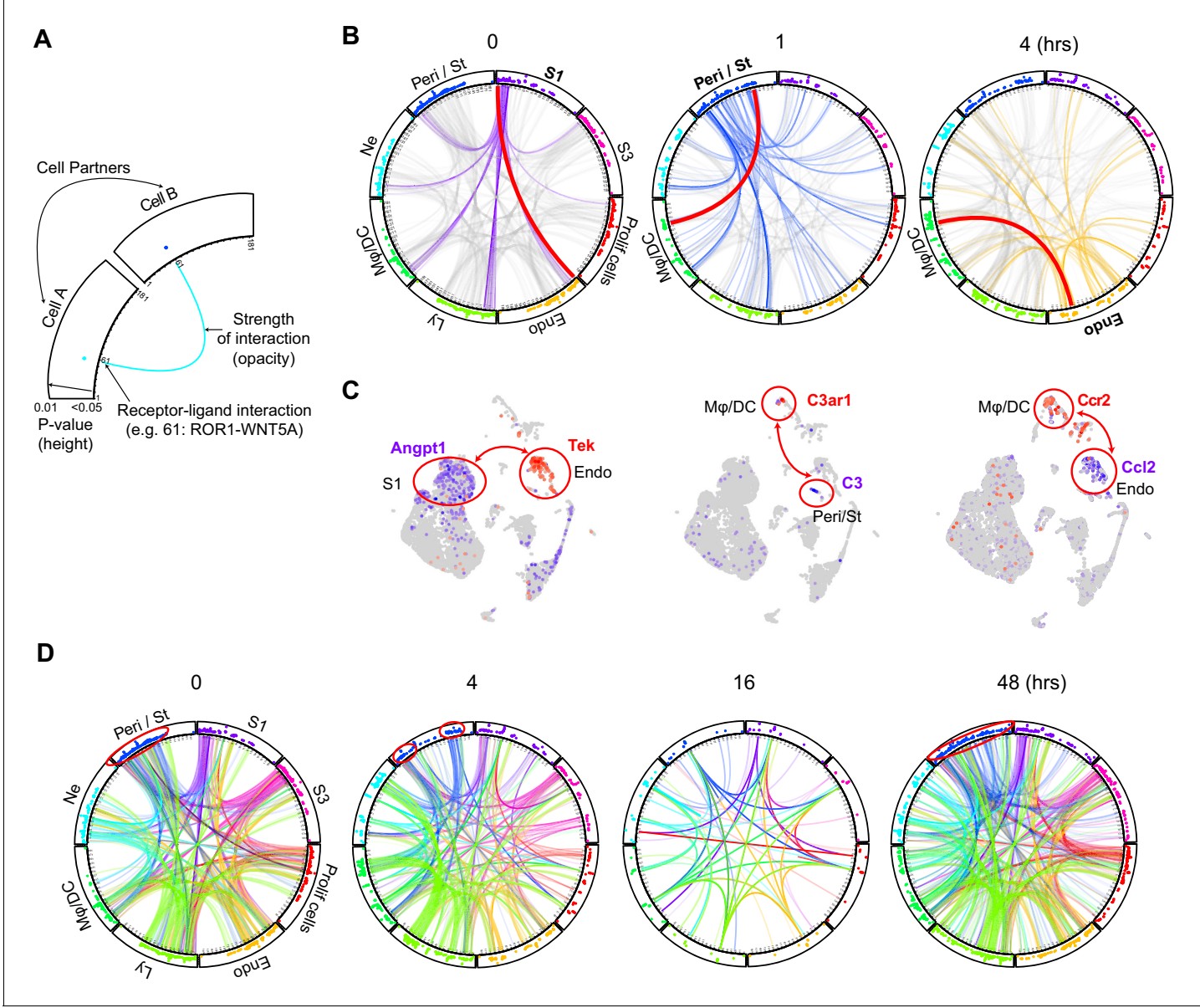

**Figure 6.** Endotoxemia alters cellular crosstalk causing time-specific global communication failure. (**A**) Schematic description of a cell–cell communication circular plot. Dots in the outer track of the circle represent specific ligands or receptors and are positioned identically for all cell types. The height of dots correlates with statistical significance (all dots are less than adjusted p-value<0.05). The identity of each dot is given in **Supplementary file 2**. (**B**) Receptor–ligand pairs for indicated cell types are displayed in circular plots. The data was generated using the CellPhone database. For clarity, communication between one cell type and all others is shown (purple lines: 0 hr for S1, blue lines: 1 hr for Peri/St and yellow lines: 4 hr for endothelial cells). Other cell–cell communications in each circular plot are shown in light gray in the background. In each circular plot, the red line connects the specific receptor–ligand pair highlighted in panel C. (**C**) Feature plots of receptor–ligand pairs between specified cell types as highlighted by the red line in panel B. In each feature plot, the ligand is shown in purple and the receptor in red. (**D**) Circular plots displaying receptor–ligand interactions between all cell types at specified time points. Examples of change in communication patterns are shown in the red circles in the outer track of the plot at 0, 4 and 48 hr. Note the dramatic drop in cell communication at 16 hr.

The online version of this article includes the following figure supplement(s) for figure 6:

**Figure supplement 1.** Expanded cell–cell communication examples.

residing between proximal tubule and macrophage/DCs may explain these complex communication patterns (*Wu et al., 2019*). Such communication patterns among these four cell types may also explain macrophage clustering around S1 tubules at later time points in endotoxemia as we previously reported (*Hato et al., 2018*).

When examined comprehensively, receptor–ligand signaling progressed from a broad pattern at baseline into a more discrete and specialized one 4 hr after LPS (*Figure 6D*). Sixteen hours after LPS, we noted a dramatic drop in cell–cell communication between all cell types (*Figure 6—figure supplement 1B*). This communication failure may contribute to the transcription and translation shutdown we recently reported at this time point (*Hato et al., 2019*). In our reversible endotoxemia model, cell–cell communication recovered by 27–48 hr.

## Global communication failure is accompanied by increased activity of genes involved in recovery

Transcription factors and their downstream targets (regulons) are important regulators of a myriad of pathways involved in the pathophysiology of sepsis. Therefore, we next examined the activity of regulons along the endotoxemia timeline in all renal cells. Surprisingly, we noted in many cell types an increase in regulon activity of key transcription factors at the 16 hr time point (*Supplementary file 3*). In S1, many of the regulons active at this time point are involved in cell differentiation, development, transcription and proliferation (*Sox4, Sox9, Hoxb7, Srf*; *Figure 7A–C*). As discussed above, this time point corresponds to translation shutdown as well as cell–cell communication failure. Therefore, this 16 hr time point is not merely a time of complete shutdown and failure of the kidney. Rather, it is also a crucial transition point where key regulators of recovery and healing are activated.

## The murine endotoxemia timeline allows staging of human sepsis

Finally, we asked whether our mouse endotoxemia timeline could be used to stratify human sepsis AKI. To this end, we selected the differentially expressed genes for each time point across the mouse endotoxemia timeline (*Supplementary file 4*). We then examined the orthologues of these defining genes in human kidney biopsies of patients with the clinical diagnosis of sepsis and AKI (*Supplementary file 5*). As shown in *Figure 7D*, our approach using the mouse data succeeded in partially stratifying the human biopsies into early, mid and late sepsis-related AKI (see also Materials and methods). The severity of clinical sepsis as determined by Sequential Organ Failure Assessment Score (SOFA) correlated well with the gene expression-based patient stratification (*Figure 7E* and *Figure 7—figure supplement 1*). For example, patients whose gene expression matched that of later time points in the murine endotoxemia timeline (e.g. recovery) had the lowest SOFA score. These findings suggest that underlying injury mechanisms are partially conserved, and the mouse timeline may be valuable in staging and defining biomarkers and therapeutics in human sepsis.

## Discussion

In this work, we provide comprehensive transcriptomic profiling of the kidney in a murine endotoxemia model. To our knowledge, this is the first description of spatial and temporal endotoxin-induced transcriptomic changes in the kidney that extend from early injury well into the recovery phase. Our data cover nearly all renal cell types and are time-anchored, thus providing a detailed and precise view of the evolution of endotoxemia in the kidney at the cellular and molecular levels.

Using a combination of analytical approaches, we identified marked phenotypic changes in multiple cell populations in response to endotoxin. The proximal tubular S1 segment exhibited significant alterations consisting of early loss of traditional function-defining markers (e.g. transporters such as SGLT2). Such deranged expression of tubular transporters may explain the frequently observed alterations in water, electrolytes, acid base, and glucose homeostasis. Furthermore, the global cell–cell communication failure likely underlies kidney shutdown that cannot be ascribed solely to hypoperfusion and ischemic injury (e.g. hyperdynamic sepsis). Concomitantly, we observed novel epithelial expression of immune-related genes such as those involved in antigen presentation. This indicates a dramatic switch in epithelial function from transport and homeostasis to immunity and defense. These phenotypic changes were reversible, underscoring the remarkable resilience and

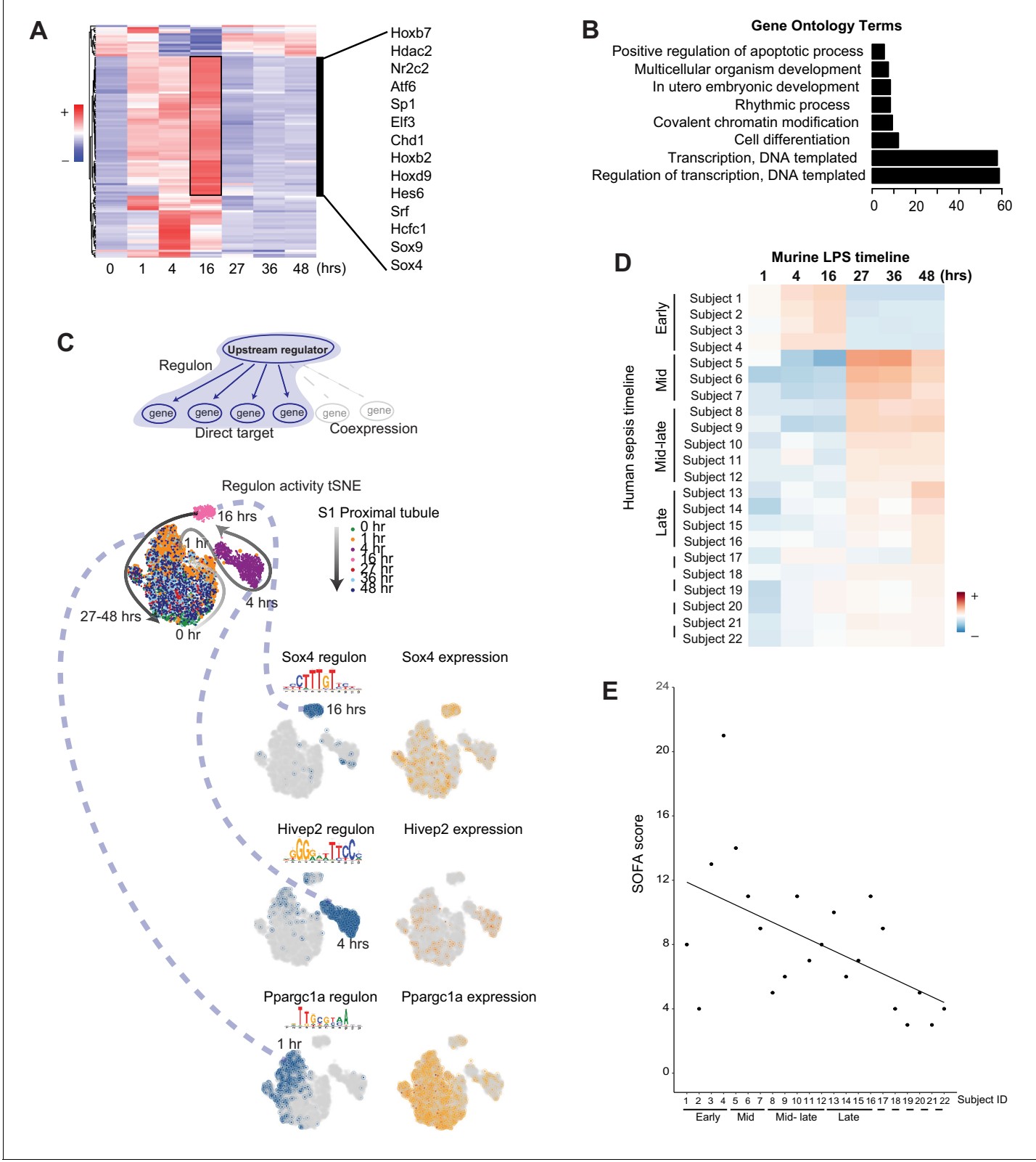

**Figure 7.** Global communication failure is accompanied by increased activity of genes involved in recovery. (**A**) SCENIC-derived heatmap of regulons for S1 tubules. Highlighted are select transcription factors with active regulons at the 16 hr time point. (**B**) Gene ontology pathway enrichment analysis derived from all regulons active at the 16 hr time point (*Supplementary file 3*). (**C**) t-SNE of proximal tubule S1 time-specific regulon activity. Select transcription factor expression (orange) and its corresponding regulon expression (blue) are shown. As shown for Sox4, note the temporal differences

*Figure 7 continued on next page*

*Figure 7 continued*

between the expression of the transcription factor itself and its regulon. (**D**) Heatmap of human sepsis kidney samples stratified based on aggregates of murine time-specific orthologues. The color scale indicates the degree of correlation based on Spearman's ρ. GO, gene ontology biological processes. (**E**) SOFA score distribution of subjects (x-axis subject ID corresponds to panel D subject ID). GO, gene ontology. SOFA, sequential organ failure assessment.

The online version of this article includes the following figure supplement(s) for figure 7:

**Figure supplement 1.** Characteristics of subjects.
**Figure supplement 2.** Serum biomarkers and chemokine/cytokine levels.

plasticity of the renal epithelium. Our data regarding loss of transport function and acquisition of immune-related functions have also been reported by others in bulk kidney tissue (*Tran et al., 2011*). Despite the significant agreement between our data and these earlier studies, some differences were noted in the time course of these changes that may be related to different dosage of endotoxin and choice of time points. Our data extends these earlier studies in bulk tissue by providing spatial resolution at the single cell level. In addition, our combined analytical tools clearly identified unique subclusters within each epithelial cell population (e.g. cortical S3 and OS S3). These subclusters likely represent cell populations that may be in part influenced by the complex microenvironments in the kidney.

Similarly, we also identified unique features in immune-cell populations. For example, the combined use of RNA velocity field and pseudotime analyses uncovered differences in macrophage subtypes relating to RNA kinetics and cell differentiation trajectories. Of note is that these subtypes only partially matched the traditional flow cytometry-based classification of macrophages (e.g. M1/M2). Therefore, the use of scRNA-seq is a powerful approach that will add to and complement our current understanding of the immune cell repertoire in the kidney.

Additional approaches such as receptor-ligand crosstalk and gene regulatory network analyses identified unique cell- and time-dependent players involved in the pathophysiology of endotoxemia. Importantly, the expression of genes involved in vectorial transport, inflammation, vascular health, coagulation and complement cascades varied greatly along the endotoxemia timeline, and required simultaneous contributions from multiple cell types. However, these complex interactions collapsed at the 16 hr time point. This indeed is a remarkable juncture in the endotoxemia timeline that we have previously investigated in other models of murine sepsis including cecal ligation and puncture/ CLP (*Hato et al., 2019*). It is the time where profound translation failure and organ shutdown occur. Our current data point to massive cell-cell communication failure as a key feature of this time point. Surprisingly, it is also at this time point that reparative pathways started to emerge. It is thus a defining point in sepsis that may have significant clinical implications. For example, the recovery process begins at the time when traditional markers such as serum creatinine is still increasing (*Figure 7—figure supplement 2*). Importantly, molecules involved in this recovery (e.g. Sox9) can be future targets at the bedside and thus have clinical potential to treat sepsis-associated AKI.

Our work points to the urgent need for defining a more accurate and precise human sepsis timeline. Such definition will guide the development of biomarkers and therapies that are cell and time specific. In our data, the gene expression changes in murine endotoxemia did partially map to tissues from septic patients, thus allowing temporal stratification of human sepsis. These precisely time- and space-anchored data will provide the community with rich and comprehensive foundations that will propel further investigations into human sepsis.

## Limitations

We recognize that endotoxemia and other murine models may not perfectly recapitulate human sepsis. Nevertheless, these models partially exhibit molecular similarities to various stages of human sepsis. Cecal ligation and puncture (CLP) may mimic human sepsis better than endotoxemia. However, CLP is a polymicrobial model that may be intractable at the molecular level. It carries very high mortality and is thus more useful for survival interventions. In contrast, endotoxemia is a well-defined ligand-receptor pathway that can be dissected with precision. With the endotoxin dose used in this study, it is also a reversible model that allows the investigation of recovery pathways. Another

limitation of our study is the use of young healthy mice thus losing the contributions of age and comorbidities to the pathophysiology of sepsis.

# Materials and methods

## Key resources table

| Reagent type (species) or resource | Designation | Source or reference | Identifiers | Additional information |
|---|---|---|---|---|
| Commercial assay, kit | RNAscope probe-Mm-Agt | Advance Cell Diagnosis | Cat. No. 426941 | |
| Commercial assay, kit | RNAscope probe-Mm-Aqp1 | Advance Cell Diagnosis | Cat. No. 504741-C2 | |
| Commercial assay, kit | Milliplex MAP Mouse Cytokine/Chemokine Magnetic Bead Panel–Premixed 32 Plex | Millipore | Cat. No. MCYTMAG-70K-PX32 | |
| Commercial assay, kit | Annexin V dead cell removal kit | Stem Cell Technologies | Cat. No. 17899 | |
| Commercial assay, kit | Multi-Tissue Dissociation Kit 2 | Miltenyi Biotec | Cat. No. 130-110-203 | |
| Commercial assay, kit | Chromium Single Cell 3' Library and gel bead kit | 10x Genomics | Cat. no. 1000121 | |
| Commercial assay, kit | NovaSeq 6000 S1 reagent kit | Illumina | Cat. No. 20012865 | |
| Commercial assay, kit | Visium Spatial Gene Expression library preparation slide | 10x Genomics | Cat. No. 1000200 | |
| Chemical compound, drug | Red blood cell lysing buffer Hybri-Max | Sigma | Cat. No. R7757 | |
| Antibody | NFkB P65 (D14E12 rabbit monoclonal) | Cell Signaling | Cat. 8242S | |
| Chemical compound, drug | CldU | Sigma | Cat. C6891 | |
| Strain, strain background (*Escherichia coli*) | LPS *E. coli* serotype o111:B4 | Sigma | Cat. No. L2630 Lot No. 095M4163V | |
| Biological samples | Human renal biopsy bulk AKI RNAseq data | | PMID:30507610 | GEO: GSE122274 |
| Software, algorithm | Monocle | *Cao et al., 2019* | PMID:30787437 | |
| Software, algorithm | Seurat | *Stuart et al., 2019*; *Butler et al., 2018* | RRID:SCR_016341 | https://satijalab.org/ |
| Software, algorithm | SCENIC | *Aibar et al., 2017* | PMID:28991892 | |
| Software, algorithm | Cellphone DB | *Efremova et al., 2020*; *Vento-Tormo et al., 2018* | | https://www.cellphonedb.org/ |
| Software, algorithm | RNA velocity | *La Manno et al., 2018* | PMID:30089906 | |
| Software, algorithm | SingleR | *Aran et al., 2019* | PMID:30643263 | |
| Software, algorithm | Harmony and Palantir | *Nowotschin et al., 2019*; *Setty et al., 2019* | PMID:30959515 PMID:30899105 | |
| Software, algorithm | R | R Project for Statistical Computing | RRID:SCR_001905 | http://www.r-project.org/ |
| Commercial protocol | RNAscope multiplex Fluorescent Reagent Kit v2 | Advance Cell Diagnosis Inc | | |

*Continued on next page*

*Continued*

| Reagent type (species) or resource | Designation | Source or reference | Identifiers | Additional information |
|---|---|---|---|---|
| Other | Dead cell removal protocol using Annexin V | https://cdn.stemcell.com/media/files /pis/DX21956-PIS_1_0_1.pdf?_ga= 2.34218465.1547447083.1547219505 %E2%80%93776976877.1534951026 | | Commercial protocol |
| Other | Chromium Single Cell 3' Reagent Kits V3 User Guide | https://assets.ctfassets.net/ an68im79xiti/51xGuiJhVKOeIIceW88gsQ/ 1db2c9b5c9283d183ff4599fb489a720/ CG000183_ChromiumSingleCell3 __v3_UG_Rev-A.pdf | | Commercial protocol |
| Other | Dissociation of mouse kidney using the Multi Tissue Dissociation Kit 2 | https://www.miltenyibiotec.com/ upload/assets/IM0015569.PDF | | Commercial protocol |

## Experimental model and subject details

Animal model: Male C57BL/6J mice were obtained from the Jackson Laboratory. Mice were 8–10 weeks of age and weighed 20–25 g. They were subjected to a single dose of 5 mg/kg LPS tail vein injection *E. coli* serotype 0111:B4 Sigma. This dose of endotoxin carries no mortality and results in rapid induction of cytokines/chemokines and causes *reversible AKI* as shown by serum creatinine (*Figure 7—figure supplement 2*). Animals were sacrificed at 0, 1, 4, 16, 27, 36, and 48 hr after LPS (both kidneys per animal for each time point).

## Study approval

All animal protocols were approved by the Indiana University Institutional Animal Care Committee and conform to the NIH (*Guide for the Care and Use of Laboratory Animals*, National Academies Press, 2011). The study in humans was approved by the Indiana University Institutional Review Board (protocol no. 1601431846). As only archived human biopsies were used in this study, the Institutional Review Board determined that informed consent was not required.

## Isolation of single cell homogenate from murine kidneys

Murine kidneys were transported in RPMI1640 (Corning), on ice immediately after surgical procurement. Kidneys were rinsed with PBS and minced into eight sections. Each sample was then enzymatically and mechanically digested for 30 min (Multi-Tissue Dissociation Kit two and GentleMACS dissociator/tube rotator, Miltenyi Biotec). The samples were prepared per protocol 'Dissociation of mouse kidney using the Multi Tissue Dissociation Kit 2' with the following modifications: After termination of the program 'Multi_E_2', we added 10 mL RPMI1640 (Corning) and 10% BSA (Sigma-Aldrich) to the mixture, filtered and homogenate was centrifuged (300 g for 5 min at 4°C). The cell pellet was resuspended in 1 mL of RBC lysis buffer (Sigma), incubated on ice for 3 min, and cell pellet washed three times (300 g for 5 min at 4°C). Annexin V dead cell removal (Stem Cell Technologies) was performed using magnetic bead separation after final wash, and the pellet resuspended in RPMI1640/BSA 0.04%. Viability and counts were assessed using Trypan blue (Gibco) and brought to a final concentration of 1 million cells/mL, exceeding 80% viability as specified by 10x Genomics processing platform.

## Single cell library preparation

The sample was targeted to 10,000 cell recovery and applied to a single cell master mix with lysis buffer and reverse transcription reagents, following the Chromium Single Cell 3' Reagent Kits V3 User Guide, CG000183 Rev A (10X Genomics). This was followed by cDNA synthesis and library preparation. All libraries were sequenced in Illumina NovaSeq6000 platform in paired-end mode (28 bp + 91 bp). Fifty thousand reads per cell were generated and 91% of the sequencing reads reached Q30 (99.9% base call accuracy). The total number of recovered cells for all time points was 63,287 cells, and per experiment was 9191 (0 hr), 9460 (1 hr), 9865 (4 hr), 5165 (16 hr), 7678 (27 hr), 10,119 (36 hr), and 11,809 (48 hr after LPS).

## Single cell data processing

The 10x Genomics Cellranger (v. 2.1.0) pipeline was utilized to demultiplex raw base call files to FASTQ files and reads aligned to the mm10 murine genome using STAR (*Dobin et al., 2013*). Cellranger computational output was then analyzed in R (v.3.5.0) using the Seurat package v. 3.0.1 (*Stuart et al., 2019*). Seurat objects were created for non-integrated and integrated (inclusive of all time points) using the following filtering metrics: gene counts were set between 200–3000 and mitochondrial gene percentages less than 50 to exclude doublets and poor quality cells. Gene counts were log transformed and scaled to $10^4$. The top 20 principal components were used to perform unsupervised clustering analysis, and visualized using UMAP dimensionality reduction (resolution 1.0). Using the Seurat package, annotation and grouping of clusters to cell type was performed manually by inspection of differentially expressed genes (DEGs) for each cluster, based on canonical marker genes in the literature (*Kretzler and Ju, 2015*; *Lake et al., 2019*; *Lee et al., 2015*; *Park et al., 2018*; *Wu et al., 2018*). In some experiments, we used edgeR negative binomial regression to model gene counts and performed differential gene expression and pathway enrichment analyses (topKEGG, topGO, *Figure 5*, *Figure 5—figure supplement 1A*, *Figure 5—figure supplement 2*, and DAVID 6.8, *Figure 7B*). (*Alessandrì et al., 2019*; *Huang et al., 2009*).

The immune cell subset was derived from the filtered, integrated Seurat object. UMAP resolution was set to 0.4, which yielded 14 clusters. The clusters were manually assigned based on inspection of DEGs for each cluster, and cells grouped if canonical markers were biologically redundant. We confirmed manual labeling with an automated labeling program in R, SingleR (*Aran et al., 2019*).

To compare our scRNA-seq data against publicly available bulk kidney array data (GSE30576), affymetrix ID was converted to Gene Name and those without a match were discarded. edgeRs glmQLFTest was used to generate comparisons for each individual one-to-one group comparison within the samples. In scRNA-seq data, pseudobulk kidneys were generated by randomly selecting 2000 cells from each condition as a group (seed 999) followed by one-to-one group comparison. The output of these comparisons were DE gene lists which were sorted by FDR. These lists were split into those in which the logFC was positive or negative and the top 500 genes were extracted. Jaccard heatmaps were generated by using the gene lists from each condition comparison to find the number of shared genes for each condition and generating a Jaccard index using hclust complete method.

## Upstream regulatory network analysis

SCENIC analysis (*Aibar et al., 2017*) was performed using the default setting and mm9-500bp-upstream-7species.mc9nr.feather database was used for data display.

## Pseudotemporal ordering of single cells

We performed pseudotime analysis on the integrated Seurat object containing all cell types as well as the immune cell subset. Cells from each of the seven time points were included and were split into individual gene expression data files organized by previously defined cell type. These individual datasets were analyzed separately through the R package Monocle using default parameters. Outputs were obtained detailing the pseudotime cell distributions for each cell type. Positional information for the monocle plot was used to subset and color cells for downstream analyses (*Trapnell et al., 2014*). We performed a separate temporal ordering analysis of S1, S2, and S3 proximal tubule segments across all time points and visualized using t-SNE, produced by Harmony and Palantir R packages (*Setty et al., 2019*).

## RNA velocity analysis

BAM files were fed through the velocyto pipeline (*La Manno et al., 2018*) to obtain. loom files for each experimental condition. These loom files along with their associated UMAP positions and principal component tables extracted from the merged Seurat file were then fed individually into the RNA Velocity pipeline as described in the Velocyto.R Dentate Gyrus/loom tutorial. The default settings described in the tutorial were used except for tSNE positions that were overwritten with the associated UMAP positions from the merged Seurat object, as well as the principal component table. This generated an RNA velocity figure mapped using the merged Seurat object cell positions. Similar analysis was done for the immune subsetted data.

## Cell–cell communication analysis

We applied the Cellphone database (*Efremova et al., 2020*) of known receptor-ligand pairs to assess cell–cell communication in our integrated dataset. Gene expression data from the integrated Seurat file was split by time point and genes renamed to Human gene names then reformatted into the input format described on the CellphoneDB website. Individual time point samples were fed into the web document on the cellphone dB website using 50 iterations, precision of 3, and 0.1 ratio of cells in a cluster expressing a gene. Output files for each time point obtained from the website were merged, then interactions trimmed based on significant sites and only selecting secreted interactions.

To visualize cellular cross talk, we applied this data to a circular plot. The interactions from the merged, trimmed cellphone dB file were sorted by cluster interaction then consolidated into 17 final cell types. Each cell type contained a list of significant interacting pairs (with $p < 0.05$) and their associated strength values (the larger the value the smaller the p value). These were then visualized using R Circlize package (*Gu et al., 2014*).

## Human sepsis staging

To determine time-defining murine DEGs, we randomly selected 2000 cells for every time point from all clusters and normalized the data using edgeR function calcNormFactors. DEGs were determined between one time point versus all others and significant genes filtered by selecting for FDR < 0.05 (*Supplementary file 4*). Human specimens were derived from OCT cores of kidney biopsy or nephrectomy samples (GSE139061). All biopsy specimens (N = 22) had a primary pathology diagnosis of AKI and were acquired in clinical care of patients with a diagnosis of sepsis (*Hato et al., 2019*). Some of the reference nephrectomies were obtained from University of Michigan through the Kidney Precision Medicine Project. These reference nephrectomies were from unaffected portions of tumor nephrectomies or deceased donors. A bulk 20 µm cross-section was cut from each OCT core and RNA was extracted using the Arcturus Picopure extraction kit (KIT0214). Libraries were prepared with the Takara SMARTer Stranded Total RNA-Seq Kit v2 Pico Input. Sequencing was performed on an Illumina HiSeq 4000. The murine genes from each time point were translated to their respective human orthologues using the biomaRt package and ensembl database. Each gene had its expression fold change calculated for each time point in relation to all other time points in the mouse. Separately for each human biopsy specimen, the expression of each gene was calculated as a fold change compared to the mean of all reference samples (*Supplementary file 5*). A spearman correlation assessed alignment between the fold changes of the mouse and human data. Data were displayed as a heatmap. The genes used to generate the heatmap are shown in *Supplementary file 6*.

## Spatial transcriptomics

A mouse kidney was immediately frozen in Optimal Cutting Temperature media. A 10 µm frozen tissue section was cut and affixed to a Visium Spatial Gene Expression library preparation slide (10X Genomics). The specimen was fixed in methanol and stained with hematoxylin-eosin reagents. Images of hematoxylin-eosin-labeled tissues were collected as mosaics of 10x fields using a Keyence BZ-X810 fluorescence microscope equipped with a Nikon 10X CFI Plan Fluor objective. The tissue was then permeabilized for 12 min and RNA was isolated. The cDNA libraries were prepared and then sequenced on an Illumina NovaSeq 6000. Using Seurat 3.1.4, we identified anchors between the integrated single cell object and the spatial transcriptomics datasets and used those to transfer the cluster data from the single cell to the spatial transcriptomics. For each spatial transcriptomics spot, this transfer assigns a score to each single cell cluster. We selected the cluster with the highest score in each spot to represent its single cell associated cluster. Using a Loupe Browser, expression data was visualized overlying the hematoxylin-eosin image.

## Single-molecule RNA in situ hybridization

Formalin-fixed paraffin-embedded cross sections were prepared with a thickness of 5 µm. The slides were baked for 60 min at 60℃. Tissues were incubated with Xylene for 5 min x2, 100% ETOH for 2 min x2, and dried at room temperature. RNA in situ hybridization was performed using RNAscope multiplex Fluorescent Reagent Kit v2 (Advance Cell Diagnosis Inc) as per the manufacturer

instructions. Probe sets were obtained from Advance Cell Diagnosis Inc (murine Agt Cat. No. 426941, Aqp1 Cat. No. 504741-C2). TSA Cyanine 3 Plus and Fluorescein Plus Evaluation kit (PerkinElmer, Inc) was used as secondary probes for the detection of RNA signals. All slides were counterstained with DAPI and coverslips were mounted using fluorescent mounting media (ProLong Gold Antifade Reagent, Life Technologies). The images were collected with a LSM800 confocal microscope (Carl Zeiss).

### In vivo thiamine analog labeling
Animals were injected with CldU (Sigma) 50 µg/g i.p. twice before harvesting the kidney tissues (−12 hr and −2 hr). Fixed tissues were deparaffinized and stained with rat anti-BrdU antibody, which binds to CldU (BU1/72, Abcam ab6326).

### Cytokine/chemokine multiplex
Analysis of serum and kidney homogenate cytokines/chemokines was performed using Milliplex MAP Mouse Cytokine/Chemokine Magnetic Bead Panel–Premixed 32 Plex (MCYTMAG-70K-PX32; Millipore). Kidney tissue proteins were extracted with RIPA buffer, and total protein concentrations were adjusted to 0.4 µg/µl per well according to the manufacturer's instructions. Analysis was performed by the Multiplex Analysis Core at Indiana University.

### Immunohistochemistry
Kidney tissues were fixed with 4% paraformaldehyde, deparaffinized and 5 µm sections cut. Low pH antigen retrieval was performed and tissue was stained for NF-κB p65 (D14E12 rabbit monoclonal, 1:800 dilution, Cell Signaling 8242S), envision+ rabbit DAB chromogen detection system (Dako) was used. Images were collected using Keyence BZ-X810 microscope.

### Quantification and statistical analysis
No blinding was used for animal experiments. All data were analyzed using R software packages, with relevant statistics described in results, methods and figure legends.

### Data availability
The scRNA-seq data and Visium spatial transcriptomics data were deposited in the NCBI's Gene Expression Omnibus database (GEO GSE151658, GSE154107).

Scripts are available through GitHub: https://github.com/hato-lab/kidney-endotoxin-sepsis-time-line-featureplot (*McCarthy, 2020*; copy archived at swh:1:rev: 2e4dde0759965ce51220bdb5d76dcd4da0c528be) and https://github.com/hato-lab/kidney-endo-toxin-sepsis-timeline-CellphoneDB-CirclePlot (*Myslinski, 2020*; copy archived at swh:1:rev: b2e0e84daaae3846d2f2eaa57376080fee8954f9).

## Acknowledgements
We thank the Kidney Precision Medicine Project for making data available from human kidney reference nephrectomy specimens. We thank Daria Barwinska, Constance Temm and Connor Gulbronson for assistance with specimen validation, tissue staining and imaging. Measurement of serum creatinine concentration was performed by John Moore, Yang Yan, et al. at the University of Alabama at Birmingham/UCSD O'Brien Center Core for Acute Kidney Injury Research (NIH P30DK079337) using isotope dilution liquid chromatography–tandem mass spectrometry. This work was supported by NIH K08-DK113223 and R01-AI148282 to TH, NIH R01-DK080063 and Veterans Affairs Merit (1I01B × 002901) to PCD, K08-DK107864 to MTE, the Indiana Clinical and Translational Sciences Institute (UL1TR002529), T32HL091816 and T32DK120524 to DJ.

## Additional information

### Funding

| Funder | Grant reference number | Author |
|---|---|---|
| NIH Office of the Director | K08-DK113223 | Takashi Hato |
| NIH Office of the Director | R01-DK080063 | Pierre C Dagher |
| NIH Office of the Director | K08-DK107864 | Michael T Eadon |
| U.S. Department of Veterans Affairs | 1I01BX002901 | Pierre C Dagher |
| Indiana Clinical and Translational Sciences Institute | UL1TR002529 | Danielle Janosevic |
| NIH Office of the Director | T32DK120524 | Danielle Janosevic |
| NIH Office of the Director | R01-AI148282 | Takashi Hato |
| Indiana Clinical and Translational Sciences Institute | T32HL091816 | Danielle Janosevic |

The funders had no role in study design, data collection and interpretation, or the decision to submit the work for publication.

### Author contributions

Danielle Janosevic, Data curation, Formal analysis, Funding acquisition, Investigation, Visualization, Methodology, Writing - original draft, Writing - review and editing; Jered Myslinski, Formal analysis, Visualization; Thomas W McCarthy, Software, Formal analysis, Visualization; Amy Zollman, Data curation, Investigation; Farooq Syed, Investigation; Xiaoling Xuei, Hongyu Gao, Kimberly S Collins, Ying-Hua Cheng, Seth Winfree, Tarek M El-Achkar, Bernhard Maier, Data curation; Yun-Long Liu, Data curation, Methodology; Ricardo Melo Ferreira, Formal analysis; Michael T Eadon, Data curation, Funding acquisition; Takashi Hato, Conceptualization, Resources, Data curation, Software, Formal analysis, Supervision, Funding acquisition, Validation, Investigation, Visualization, Methodology, Writing - original draft, Project administration, Writing - review and editing; Pierre C Dagher, Supervision, Funding acquisition, Writing - original draft, Project administration, Writing - review and editing

### Author ORCIDs

Danielle Janosevic (iD) https://orcid.org/0000-0003-3215-6942
Thomas W McCarthy (iD) http://orcid.org/0000-0002-7734-2821
Farooq Syed (iD) http://orcid.org/0000-0002-0284-0631
Bernhard Maier (iD) http://orcid.org/0000-0002-8174-0873
Ricardo Melo Ferreira (iD) http://orcid.org/0000-0003-2063-9744
Takashi Hato (iD) https://orcid.org/0000-0002-0446-6575

### Ethics

Human subjects: The study in humans was approved by the Indiana University Institutional Review Board (protocol no. 1601431846). As only archived human biopsies were used in this study, the Institutional Review Board determined that informed consent was not required.
Animal experimentation: All animal protocols were approved by the Indiana University Institutional Animal Care Committee and conform to the NIH (Guide for the Care and Use of Laboratory Animals, National Academies Press, 2011).

### Decision letter and Author response

Decision letter https://doi.org/10.7554/eLife.62270.sa1
Author response https://doi.org/10.7554/eLife.62270.sa2

## Additional files

### Supplementary files
- Supplementary file 1. Cluster-defining markers.
- Supplementary file 2. Receptor-ligand interaction.
- Supplementary file 3. SCENIC regulon activity.
- Supplementary file 4. Differentially expressed genes for each time point across the mouse endotoxemia timeline.
- Supplementary file 5. Human kidney biopsy data.
- Supplementary file 6. Human kidney biopsy data: genes used for generating *Figure 7D* heatmap.
- Transparent reporting form

### Data availability

The scRNA-seq data and spatial transcriptomics data have been deposited in the NCBI's Gene Expression Omnibus database (GEO GSE151658, GSE154107). We also provide interactive websites: https://connect.rstudio.iu.edu/content/18/ https://connect.rstudio.iu.edu/content/19/ Scripts are available through GitHub: https://github.com/hato-lab/kidney-endotoxin-sepsis-timeline-featureplot (copy archived at https://archive.softwareheritage.org/swh:1:rev:2e4dde0759965-ce51220bdb5d76dcd4da0c528be/) and https://github.com/hato-lab/kidney-endotoxin-sepsis-time-line-CellphoneDB-CirclePlot (copy archived at https://archive.softwareheritage.org/swh:1:rev:b2e0e84daaae3846d2f2eaa57376080fee8954f9/).

The following datasets were generated:

| Author(s) | Year | Dataset title | Dataset URL | Database and Identifier |
|---|---|---|---|---|
| Janosevic D, Hato T, McCarthy T | 2020 | The orchestrated cellular and molecular responses of the kidney to endotoxin define a precise sepsis timeline | https://www.ncbi.nlm.nih.gov/geo/query/acc.cgi?acc=GSE151658 | NCBI Gene Expression Omnibus, GSE151658 |
| Eadon MT, Hato T, Ferreira RM, Janosevic D | 2020 | The orchestrated cellular and molecular responses of the kidney to endotoxin define a precise sepsis timeline | https://www.ncbi.nlm.nih.gov/geo/query/acc.cgi?acc=GSE154107 | NCBI Gene Expression Omnibus, GSE154107 |

The following previously published dataset was used:

| Author(s) | Year | Dataset title | Dataset URL | Database and Identifier |
|---|---|---|---|---|
| Eadon M | 2019 | Transcriptomic signatures of kidney injury in human renal biopsy specimens | https://www.ncbi.nlm.nih.gov/geo/query/acc.cgi?acc=GSE139061 | NCBI Gene Expression Omnibus, GSE139061 |

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
