## [Decision Letter]

**Acceptance summary:**

This study is deconvoluting the cellular changes that occur in the kidney in terms of kind and function in mice challenged with bacterial endotoxin. This model is a proxy for sepsis, a most serious clinical infectious condition. The investigators have used single cell sequencing and did compare their findings to those seen in renal biopsies of patients with kidney injury in sepsis. The study provides new data on the changes in immunological, endothelial and parenchymal cells.

**Decision letter after peer review:**

Thank you for submitting your article "The orchestrated cellular and molecular responses of the kidney to endotoxin define a precise sepsis timeline" for consideration by *eLife*. Your article has been reviewed by three peer reviewers, including Evangelos J Giamarellos-Bourboulis as the Reviewing Editor and Reviewer #1, and the evaluation has been overseen by Jos van der Meer as the Senior Editor. The reviewers have agreed to reveal their identity: Jesus Bermejo-Martin (Reviewer #2); Sebastian Weis (Reviewer #3).

The reviewers have discussed the reviews with one another and the Reviewing Editor has drafted this decision to help you prepare a revised submission.

Summary:

This is an elegant study deconvoluting the cellular changes in the kidney (kind and function) along the course of a sepsis-like disease model on mice challenged with LPS. For achieving this objective, the authors have employed a single cell sequencing approach, and compared their findings to those observed in renal biopsies from patients suffering from AKI secondary to sepsis. This work provides interesting new data on the changes on immunological, endothelial and parenchymal cells composition and function following the LPS challenge, revealing important new insight on the dynamics and nature of this changes.

Essential revisions:

1) One main limitation is that it is not possible to fully understand the storyline of the authors which becomes confusing at some points. Several parts of the Results start with the verb "Note" as if they are figures and not text. It is also very difficult to understand what the need of archived renal tissue from humans was and the participation of this analysis in the storyline.

2) The authors do not provide explanation on the purification of cell populations and on the methods, they used to avoid cell contamination.

3) We would like to see in the Discussion the comparison of the data of single cell RNA analysis with previously published array data.

4)The authors work with a model of LPS injection. This is all fine and can be done and provides useful information. However, it explicitly does not reflect the pathophysiology of sepsis that is defined as a deregulated host response to infection. Therefore, any reliable sepsis model needs to apply pathogens, which is not done. What we here look at, is TLR4 activation by LPS as a model of inflammatory stress. This needs to be changed. The best way would be by using an additional animal sepsis model confirming the LPS data. As an alternative, the manuscript should be rewritten in a way that it is evident, that the model is a pure TLR4 activation model of inflammatory stress and not sepsis. The discussion should be adopted accordingly. As such a short term "initial inflammatory burst" will be seen in an LPS model but would hardly reflect clinical reality.

5) In the same vein, there are additional problems concerning animal models in sepsis that need to be discussed here: for example, patients suffering from sepsis use to be aged individuals suffering from chronic conditions such as diabetes, hypertension, etc. Ageing and chronic diseases impact on the host immune system and tissues. The authors seem to employ mice with no prior comorbidities. Please address this limitation in the Discussion.

6) The authors should clarify how many mice were used to obtain the SCS results. A sentence in the Materials and methods, Results and Abstract would help. Does "mice" mean two, or more? This could help to assess data validity.

7) A correlate is missing between SCS findings and clinical / biological parameters obtained in parallel from the mice: creatinine, inflammatory proteins / cytokines, urine volume, etc.

8) The authors should provide disease severity, mortality, organ damage markers, cytokine levels for their animal model so that the reader can better assess the severity of the model.

9) Is there any data on the consequences of the different states with regard to kidney function. This would be very interesting to see.

10) The authors state that kidneys from septic patients have been used. Please provide clinical characteristics (SOFA; source of infection; microbiology, main clinical parameters; age, gender, comorbidity etc.) of these patients and of the controls. Were they really patients with sepsis that received a kidney biopsy? This is rarely done. Please explain.

11) It is unclear, why the authors swing for some lines to SARS-CoV-2. This is not the topic of the manuscript. LPS has no role in SARS-CoV-2 and no viral infections are being studies here. This should be removed.

12) As the authors provide IHC data (Figure 1—figure supplement 2) and as there is a LPS directly regulates NF-κB function, we would suggest to specifically add data for NF-κB activation with IHC.

[Editors' note: further revisions were suggested prior to acceptance, as described below.]

Thank you for submitting your work entitled "The orchestrated cellular and molecular responses of the kidney to endotoxin define a precise sepsis timeline" for consideration by *eLife*. Your article has been reviewed by a Senior Editor, a Reviewing Editor, and three reviewers. The reviewers have opted to remain anonymous.

Our decision has been reached after consultation between the reviewers. Based on these discussions and the individual reviews below, we regret to inform you that your work will not be considered further for publication in *eLife*.

The main reasons for rejection are that the reviewers and the reviewing editor feel that you ignored the advice as given in the reviews. Not only our strong advice regarding the phrasing of sepsis and the LPS challenge was not followed, but other issues were not addressed (see the review below).

In principle eLife only allows one round of revision, but if you feel you could fully comply with our criticism and send us a revised paper within two weeks, we would be willing to reconsider our decision.

Reviewer #3:

Janosevic et al., submit the first revision of their manuscript entitled "The orchestrated cellular and molecular responses of the kidney to endotoxin define a precise sepsis timeline". For the general assessment I kindly refer to my first review.

The first joint review was rather positive and encouraged the submission of an adapted version of the manuscript. The authors however have only partly included the reviews recommendations despite some clear indications what would need to be changed.

In more detail.

i) The authors do not discuss their single cell RNA data with previously published data, as requested.

ii) Although that the reviewer stated specifically that the presented work is not sepsis and ask for a change, this was not followed convincingly in the revised version. In the point-to-point reply two references are given. Yet the two well-known and contradicting papers (from the same datasets) by Seok and the Takao study do not investigate the correlation between sepsis and endotoxin application. Instead, the comparison is focused on the comparability of human and murine models with opposing results. At least for me, this does not support the notion to use LPS as a model of sepsis.

iii) The issue that sepsis patients are likely to have older age and comorbidities which is not the case in the young mice are not being discussed as a limitation, as requested.

iv) the clinical data is sparse. If patients' samples are being used, the authors should be able to retrieve the requested information.

v) Data for NfKB activation using IHC is not provided.

vi) Consequences with regard to kidney function are not discussed.

vii) Disease severity/mortality is not provided as requested.

It is unclear to me and unfortunate why the authors decided so, however, as the critics from the first round stand, I cannot recommend it for publication at the current stage.

---

## [Author Response]

Essential revisions:1) One main limitation is that it is not possible to fully understand the storyline of the authors which becomes confusing at some points. Several parts of the Results start with the verb "Note" as if they are figures and not text. It is also very difficult to understand what the need of archived renal tissue from humans was and the participation of this analysis in the storyline.

Our findings show space- and time-dependent changes in the kidney following endotoxin challenge with an organized participation of various cell populations. This has relevance to the design and administration of therapy to precisely target unique cell populations at specific time points. We also showed phenotypic changes such as loss of transporters that can alter the function of the renal epithelium and may explain some clinical findings in septic patients. Furthermore, we showed that all these changes culminate in cell-cell communication failure at exactly the same time point characterized by translation shutdown as we have previously shown. Importantly, several genes and pathways involved in tissue healing become activated at this time point, indicating that it is a time of transition from injury to recovery.

As to the archived renal tissue, it was obtained from patients with the diagnosis of acute renal failure in the setting of sepsis without specific chronological information as to the timing of sepsis. We show that time-specific gene expression changes in the mouse sepsis timeline can be partially mapped to some of the human samples. This allowed a staging of the human samples from early to late phases of sepsis. We believe this may have important therapeutic implications.

2) The authors do not provide explanation on the purification of cell populations and on the methods, they used to avoid cell contamination.

The single-cell RNA-seq procedure used in this paper attempted to capture all cell populations in the kidney. Therefore, the tissue dissociation step was not followed by flow sorting or other attempts to enrich any specific cell types. Cell populations thereafter were identified and grouped based on gene expression.

3) We would like to see in the Discussion the comparison of the data of single cell RNA analysis with previously published array data.

Our scRNA-seq data were well in line with published array data from bulk kidney tissue (Tran et al., 2011). For example, the changes in the thick ascending loop sodium/potassium/chloride transporter (Slc12a1/NKCC2), proximal tubular aquaporin (Aqp1), and genes involved in antigen presentation (H2-Ab1, Cd74) were comparable to those reported in bulk tissue as shown in Author response image 1. scRNA-seq has the advantage of localizing these changes to specific cell populations.

**Author response image 1. respfig1:** Bulk kidney microarray data.

4) The authors work with a model of LPS injection. This is all fine and can be done and provides useful information. However, it explicitly does not reflect the pathophysiology of sepsis that is defined as a deregulated host response to infection. Therefore, any reliable sepsis model needs to apply pathogens, which is not done. What we here look at, is TLR4 activation by LPS as a model of inflammatory stress. This needs to be changed. The best way would be by using an additional animal sepsis model confirming the LPS data. As an alternative, the manuscript should be rewritten in a way that it is evident, that the model is a pure TLR4 activation model of inflammatory stress and not sepsis. The discussion should be adopted accordingly. As such a short term "initial inflammatory burst" will be seen in an LPS model but would hardly reflect clinical reality.

We agree with the reviewers that LPS/endotoxemia model may not capture the full spectrum of Gram-negative sepsis in humans. However, as we and others have shown, it does recapitulate several key events in the sepsis timeline, and correlates well with the CLP model (Seok et al., 2013; Takao et al., 2015). As the reviewers’ state, it is a precise model that follows signaling events initiated by endotoxin/TLR4 interaction. It is a highly reproducible model that is also reversible, and therefore can be used to identify pathways involved in tissue recovery and healing. In fact, murine models of endotoxemia and its variants have been found to have significant similarities in gene expression changes to human sepsis. This has been elegantly demonstrated in Takao et al., 2015. This is also supported in our data by the successful mapping of gene expression changes in the endotoxemic mouse to human tissues from septic patients (Figure 7D). We have now changed the word “sepsis” to endotoxemia in several parts of the manuscript.

5) In the same vein, there are additional problems concerning animal models in sepsis that need to be discussed here: for example, patients suffering from sepsis use to be aged individuals suffering from chronic conditions such as diabetes, hypertension, etc. Ageing and chronic diseases impact on the host immune system and tissues. The authors seem to employ mice with no prior comorbidities. Please address this limitation in the Discussion.

We agree with the reviewers and are aware of such models. For example, one such model uses CLP in aged rats or mice. This model was originally introduced with the goal of inducing measurable renal injury which is not always apparent in traditional CLP (Doi et al., 2009). These models are primarily used to test the effect of therapeutics on renal function and animal survival. However, they pose more challenges in identifying specific and canonical molecular pathways because of their polymicrobial nature, comorbidities, and hemodynamic effects.

6) The authors should clarify how many mice were used to obtain the SCS results. A sentence in the Materials and methods, Results and Abstract would help. Does "mice" mean two, or more? This could help to assess data validity.

We pool kidneys from 3 mice per time point. The pooled 3 kidneys are run as one sample for cost purposes.

7) A correlate is missing between SCS findings and clinical / biological parameters obtained in parallel from the mice: creatinine, inflammatory proteins / cytokines, urine volume, etc.

Please see below (response to #8).

8) The authors should provide disease severity, mortality, organ damage markers, cytokine levels for their animal model so that the reader can better assess the severity of the model.

We have previously characterized this endotoxemia model extensively, including parameters such as survival, reversibility, renal function (serum creatinine, BUN), tubular injury (KIM1, NGAL), as well as tissue and serum cytokines (Kalakeche et al., 2011; Hato et al., 2015; Hato et al., 2017; Hato et al., 2018; Hato et al., 2019). The dose of endotoxin used in this model does not result in significant hemodynamic alterations (Nakano et al., 2015). We now provide additional new data on such clinical/biological parameters that characterize this murine model (New Figure 7—figure supplement 1).

9) Is there any data on the consequences of the different states with regard to kidney function. This would be very interesting to see.

Kidney tissue from septic patients typically does not show significant morphological changes that can explain the observed profound organ shutdown. Furthermore, organ shutdown in septic patients occurs even in hyperdynamic states where tissue perfusion is not compromised. Similarly, animal models of sepsis such as endotoxemia or CLP do not exhibit marked morphological changes in the kidney.

Therefore, we propose that the molecular changes reported in this paper (e.g. dysregulated translation and cell-cell communication failure) may underlie the observed organ failure. Furthermore, some of the tubular phenotypic changes we report e.g. loss of aquaporins and glucose transporters may explain dysregulated urinary concentrating mechanisms and altered glucose metabolism reported in sepsis.

10) The authors state that kidneys from septic patients have been used. Please provide clinical characteristics (SOFA; source of infection; microbiology, main clinical parameters; age, gender, comorbidity etc.) of these patients and of the controls. Were they really patients with sepsis that received a kidney biopsy? This is rarely done. Please explain.

We agree with the reviewers. The human biopsies used in this paper were not research or protocol biopsies. In fact, these are leftover tissues stored and typically discarded at later times by the pathologist. We made every effort to retrieve any available clinical data associated with each biopsy. Such clinical data from chart reviews were frequently scarce and incomplete. Furthermore, some of these biopsies were from other institutions and no detailed clinical data were forwarded to our pathologist with the biopsies. Nevertheless, we have now included all available data such as SOFA scores and microbiology when available (New Supplementary file 5). We also agree that typically kidney biopsies are not performed in septic patients. The indications for biopsy in our cohort were variable and included delayed recovery, distinguishing interstitial nephritis and tubular necrosis, or exclusion of underlying glomerulonephritis. The final diagnosis in all these biopsies was acute tubular necrosis in the setting of sepsis.

11) It is unclear, why the authors swing for some lines to SARS-CoV-2. This is not the topic of the manuscript. LPS has no role in SARS-CoV-2 and no viral infections are being studies here. This should be removed.

We agree with the reviewer and have removed the SARS-CoV-2-related data. It was originally included because of the observed increased angiotensinogen in S3 cells, which prompted us to examine the broad expression of the renin-angiotensin system. We thought this may be of general interest due to the SARSCoV-2 pandemic, but we agree that it may not help the flow of the manuscript.

12) As the authors provide IHC data (Figure 1—figure supplement 2) and as there is a LPS directly regulates NF-κB function, we would suggest to specifically add data for NF-κB activation with IHC.

The IHC imaging data in Figure 1—figure supplement 2 was shown to specifically localize the proliferating cell cluster observed in the scRNA-seq data. The imaging confirmed that the localization of these cells in tubules versus interstitium varies along the sepsis timeline. The expression of Nf-_κ_B in the kidney in sepsis/endotoxemia has been extensively investigated (Song et al., 2019; Belen-Sanz et al., 2010; Guijarro et al., 2001). Furthermore, the expression of several components of the Nf-_κ_B pathway can be localized to specific cells along the sepsis timeline using the following link to our data (e.g. *Rela, Relb, Nfkbia, Nfkb1*): https://connect.rstudio.iu.edu/content/18/ . The activation of Nf-_κ_B is evident in endothelial and stromal cells at the 1 hour time point and then is evident in immune cells and epithelial cells at later time points. c-Jun and Fos expression levels are commonly used to identify cells exhibiting stress related to the organ dissociation procedure (Denisenko et al., 2020; van den Brink et al.,2017).

[Editors’ note: what follows is the authors’ response to the second round of review.]

Reviewer #3:Janosevic et al., submit the first revision of their manuscript entitled "The orchestrated cellular and molecular responses of the kidney to endotoxin define a precise sepsis timeline". For the general assessment I kindly refer to my first review.The first joint review was rather positive and encouraged the submission of an adapted version of the manuscript. The authors however have only partly included the reviews recommendations despite some clear indications what would need to be changed.

We would like to thank the reviewer for recognizing the strengths of the manuscript. We would also like to thank the reviewer for providing clear and practical suggestions to improve the manuscript. Finally, we apologize for our previous incomplete response to the reviewer’s comments. Indeed, we had provided in the response some of the required data. We have now added all responses, discussions and new data in the manuscript.

In more detail.i) The authors do not discuss their single cell RNA data with previously published data, as requested.

To our knowledge, our work is the first examination of endotoxin-induced kidney injury covering from very early phases (1 hour~) to recovery (48 hours) at the single-cell resolution. We searched for publicly available datasets for murine endotoxemia models, which were all done in bulk kidneys. The closest time course we could find was by Tran et al., 2011 in which 10 mg/kg LPS ip (*E. coli* serotype O111:B4) was used in C57BL/6 mice and gene expression profiles were examined at 0, 18 and 42 hours post LPS (Our endotoxemia model consists of 5 mg/kg LPS iv, *E. coli* serotype O111:B4, and gene expression profiles were examined at 0, 1, 4, 16, 27, 36 and 48 hours). As shown in new Figure 4—figure supplement 2 and discussed in the Results section, Discussion and Materials and methods we demonstrate that overall gene expression changes are similar between our study and Tran et al. We accomplished this analysis by reconstructing pseudobulk kidneys from our single-cell RNA-seq data and cross-examining time-dependent gene expression changes between the two datasets using Jaccard similarity index.

ii) Although that the reviewer stated specifically that the presented work is not sepsis and ask for a change, this was not followed convincingly in the revised version. In the point-to-point reply two references are given. Yet the two well-known and contradicting papers (from the same datasets) by Seok and the Takao study do not investigate the correlation between sepsis and endotoxin application. Instead, the comparison is focused on the comparability of human and murine models with opposing results. At least for me, this does not support the notion to use LPS as a model of sepsis.

We fully agree with the reviewer that the terms endotoxemia and sepsis should not be used interchangeably. We corrected the use of these terms; our murine model now is consistently referred to as endotoxemia, not sepsis. The term sepsis appears in the main text only when we discuss human clinical sepsis.

iii) The issue that sepsis patients are likely to have older age and comorbidities which is not the case in the young mice are not being discussed as a limitation, as requested.

We agree with the reviewer that age and other comorbidities are important factors in the pathophysiology of human sepsis. These comorbidities are not accounted for in our murine model of endotoxemia. We now dedicated a whole paragraph at the end of Discussion addressing all these limitations.

iv) the clinical data is sparse. If patients' samples are being used, the authors should be able to retrieve the requested information.

We have made a serious and honest effort to retrieve all clinical data that were available. Note that these were not research biopsies and some of them originated from outside institutions. We compiled all available data in Supplementary file 5 showing SOFA score, source of infection, microbiology, clinical context, age, gender, and comorbidity. We also provide a statistical correlation between gene-based patient stratification into early and late phases of sepsis, and their SOFA score (new Figure 7E and new Figure 7—figure supplement 1, subsection “The murine endotoxemia timeline allows staging of human sepsis”).

v) Data for NfKB activation using IHC is not provided.

We now provide NfkB IHC in new Figure 5—figure supplement 3 and subsection “Endotoxemia induces an organ-wide host defense phenotype in the kidney”. The data show nuclear translocation of NfkB as early as 1 hour after endotoxin in most cell types. These data are in line with our previously published ribo-seq data (Hato et al., 2019; specifically supplemental Figure 3 and Figure 6).

vi) Consequences with regard to kidney function are not discussed.

We now addressed possible consequences of endotoxin-induced gene expression changes on global kidney function and clinical parameters (Discussion; please also see Figure 4—figure supplement 1).

vii) Disease severity/mortality is not provided as requested.

The endotoxin model used here results in transient kidney injury. The animals invariably recover and the mortality is nil. We use this nonlethal model to elucidate molecular mechanisms of renal recovery. With regard to disease severity, serum creatinine is arguably the gold standard for defining the severity of kidney failure. We provide serum creatinine data as well as other markers of tubular injury as shown in Figure 7—figure supplement 2. Histological changes are not readily detectable (e.g. Supplemental Figure 9 in Hato et al., 2015) unless leveraging advanced molecular imaging techniques such as two-photon intravital fluorescence lifetime imaging (Hato et al., 2017). Others have worked with higher doses of LPS and examined its renal outcomes. For example, in the study by Tran et al., 2011, 10 mg/kg LPS resulted in heterogeneous renal outcomes (“The majority of mice recovered normal function by 42 hours after LPS, but some had persistent AKI”) and the survival was greater than 80%. Even with the 10 mg/kg LPS, light microscopy showed only minor to no changes in their study. We provide in subsection “Experimental Model and Subject Details*”* describing the severity and mortality of our endotoxin model. Figure 7—figure supplement 2 also includes tissue and serum cytokine levels as requested by the reviewer.

Finally, we would like to take this opportunity to emphasize the rationale behind defining 16 hrs and above as late and recovery phases in our murine model of endotoxemia. This definition is based on the single-cell RNA sequencing data in which resolution of tissue injury becomes apparent after the 16 hour time point (see Author response image 2). We have also previously demonstrated that this dose of endotoxin alters gene expression and culminates in profound protein translation shutdown in the kidney at 16 hours (Hato et al., 2019). Therefore, our molecular-based timeline is anchored around gene expression changes and protein translation shutdown, and their recovery. In contrast, standard biochemical markers (e.g. serum creatinine) resulting from altered gene expression are observed at later time points.

**Author response image 2. respfig2:** Timeline of murine LPS model. (**A**) Adapted from Figure 4. Single-cell RNA-seq data demonstrating overall time-dependent gene expression changes in proximal tubules after LPS challenge. Proximal tubules exhibit major phenotypic changes at 4 hrs (red circle) and they show the most extreme phenotype at 16 hrs (blue circle). By 27 hrs, the overall phenotype returns toward baseline, suggesting that tubules are in recovery phase (gray circles). (**B**) Changes in gene expression levels of Kidney Injury Marker 1/KIM1 (an AKI biomarker). (**C**) Serum creatinine levels exhibit known lag (Adapted from Figure 7—figure supplement 2).